# The Role of Intermolecular Interaction on Aggregation-Induced Emission Phenomenon and OLED Performance

**DOI:** 10.3390/ma15238525

**Published:** 2022-11-30

**Authors:** Patrycja Filipek, Krzysztof Karoń, Hubert Hellwig, Agata Szłapa-Kula, Michał Filapek

**Affiliations:** 1Institute of Chemistry, Faculty of Science and Technology, University of Silesia, Szkolna 9, 40-007 Katowice, Poland; 2Department of Physical Chemistry and Technology of Polymers, Silesian University of Technology, M. Strzody 9, 44-100 Gliwice, Poland

**Keywords:** aggregation-induced emission (AIE), organic light-emitting diodes (OLED), donor-π-acceptor, electrochemistry, luminescence

## Abstract

In this work, the role of intermolecular interaction on the aggregation-induced emission (AIE) phenomenon and organic light-emitting diodes’ (OLEDs) performance was investigated. During the research, a group of compounds consisting of the (-CH=C(CN)(COOR)) moiety with identical energy parameters was designed using the DFT approach and successfully synthesized. The optical, electrochemical, and aggregation-induced emission properties were studied. The aggregation-induced emission of compounds has been studied in the mixture of MeCN (as a good solvent) and water (as a poor solvent) with different water fractions ranging from 0% to 99%. Moreover, the time dependence on the AIE feature was also evaluated. Thanks to their molecular structures, almost identical behavior of these substances in dilute solutions was observed. For molecules that exhibit the strong AIE phenomenon, emission efficiency increases rapidly during aggregation. What is also very interesting is it has been shown that by introducing an appropriate substituent, one can control the degree of intermolecular interactions and “control” the length of the emitted wave. Finally, OLEDs were fabricated by the spin-coating/evaporation hybrid method. Devices showed green–blueish electroluminescence (CIE coordinates: 0.107, 0.165) with maximum luminance reaching 25 cd m^−2^ and EQE reaching 2%.

## 1. Introduction

Nowadays, organic electronics and photovoltaics are developing extremely quickly. Progress in this field is possible thanks to structure ↔ properties feedback. In each step, it is better known how the molecular structure affects their physical and chemical characteristics. This, in turn, strongly influences the final device performance. Organic light-emitting diodes (OLEDs) are extensively studied due to many factors: the ease of modification of the chemical structure of small organic compounds [1], high external quantum efficiency (EQE) [2,3], and numerous practical applications [4]. Considering the emitter structure, the best OLED performances are achieved for molecules with D-π-A architecture (D—donor, A—acceptor) [1,5]. On the other hand, from a practical point of view, molecules’ properties are typically measured in solutions, whereas here they are applied as a solid-state layer or blended [6,7,8,9,10,11,12,13,14,15,16,17,18]. Thus, the crucial task is to mitigate the aggregation-caused quenching (ACQ) phenomenon. In 2001, Tang and co-workers discovered a system in which luminogen aggregation played a constructive, instead of a destructive, role [19,20]. This behavior has been named aggregation-induced emission (AIE) and has drawn much attention recently due to its high-potential applicability in OLEDs. It is stated that AIE is caused by the restriction of intramolecular rotation (RIR) [6].

Additionally, self-assembly behavior could be (in some cases) tuned by a change in the solution concentration and type of solvent. Sekar et al. have extended knowledge in the field of this phenomenon [21,22]. They have synthesized a series of styryl compounds containing a carbazole donor core and a 2,2-dicyanovinyl moiety linked through a π bridge (i.e., having D-π-A architecture). However, what is worth noting is all of the reported compounds exhibit AIE with enhanced fluorescence intensity.

To date, compounds containing (-CH=C(CN)(COOR)) moiety have mainly been investigated as photosensitizers in dye-sensitized solar cells (DSSCs) [23,24]. However, it is known that this type of molecule exhibits high efficiency of both photo- [25] and electroluminescence [3]. A strong solvatochromic effect is also reported in detail for this group of compounds [26,27]. It is of great importance that 2,2-dicyanovinyl-1,4-dihydropyridine derivatives exhibit aggregation-induced emission (AIE), as reported recently by Li et al. [28].

The most convenient way to evaluate the AIE is to study the compound behavior in a mixture of a good solvent and a poor solvent with different ratios [6]. Our group previously synthesized and investigated nine compounds with donor-CH=C(CN)_2_ architecture, but only five of them exhibited AIE features [29]. Moreover, very recently, Fei et al. reported a nonlinear correlation of emission yields during aggregation, i.e., with a minimum of the photoluminescence intensity for the poor solvent fraction between 40% and 70% [30]. At this moment, the AIE phenomenon is worthy of investigation, which should lead to the blooming of many possible applications. The subject of this work is an investigation of the AIE properties of four synthesized compounds. Two of them (namely, compounds **1** and **2**) were mentioned earlier in the literature [31,32,33,34] but not studied in terms of the AIE phenomenon. To the best of our knowledge, the subsequent two compounds (**3** and **4**) have not been described before.

## 2. Materials and Methods

### 2.1. General Methods

All chemicals and starting materials were commercially available and were used without further purification. Solvents were distilled as per the standard methods and purged with nitrogen before use. All reactions and measurements were carried out under an argon atmosphere unless otherwise indicated. Column chromatography was carried out on Merck Silica Gel 60. Thin-layer chromatography (TLC) was performed on silica gel (Merck TLC Silica Gel 60 F_254_; Merck, Darmstadt, Germany). ^1^H NMR and ^13^C NMR spectra were recorded using a Bruker Avance Ultra Shield 400 MHz spectrometer. The peaks were referenced to the residual CDCl_3_ (7.28 and 77.04 ppm) resonances in ^1^H and ^13^C NMR spectra, respectively. UV/Vis spectra were recorded with a Hewlett–Packard model 8453 UV/Vis spectrophotometer in dichloromethane solution. Electrochemical measurements were carried out with an Eco ChemieAutolab PGSTAT128n potentiostat using glassy carbon (with diam. 2 mm) as the working electrode, while platinum coil and silver wire were used as the auxiliary and the reference electrode, respectively. Potentials are referenced with respect to ferrocene (Fc), which was used as the internal standard. Cyclic and differential pulse voltammetry experiments were conducted in a standard one-compartment cell, in CH_2_Cl_2_ (DCM; Carlo Erba, HPLC grade, Bernolsheim, France), under argon. Bu_4_NPF_6_ (Aldrich; 0.2 M, 99%; Sigma-Aldrich, St. Louis, MO, USA) was used as the supporting electrolyte. The quantum theoretical calculations were performed using density functional theory (DFT), with an exchange-correlation hybrid functional B3LYP and the basis 6-311+G for all atoms. The calculations were carried out with the use of the Gaussian 09 program. OLEDs were fabricated using the spin-coating/evaporation hybrid method. Hole injection layer (Heraeus Clevios HIL 1.3N; Heraeus Epurio, Leverkusen, Germany/Dayton, OH, USA), hole transporting layer (PVK), and emitting layer (PVK:PBD + dopant, i.e., compounds **1**–**4**) were spin-coated, whereas the electron transport layer (TPBi) and cathode (LiF/Al) were evaporated. PVK: poly (9-vinylcarbazole) (MW = 90,000, Acros Organics, Geel, Belgium/Branchburg, NJ, USA); PBD: 2-(4-biphenyl)-5-(4-tert-butylphenyl)-1,2,4-oxadiazole (TCI, Portland, OR, USA); TPBi: 1,3,5-tris(1-phenyl-1H-benzimidazol-2-yl)benzene (TCI, Portland, OR, USA); LiF (99.995%, Sigma-Aldrich, St. Louis, MO, USA); and Al wire (99.9995%, Alfa Aesar, Haverhill, MA, USA) were purchased from the companies indicated in parentheses. The overall structure of the device was ITO | HIL 1.3N (50 nm) | PVK (10 nm) | PVK:PBD (50:50) with 5 mol% of **1**–**4** (app 70 nm) | TPBi (50 nm) | LiF (1 nm) | Al (100 nm). OLED devices were fabricated using pre-cleaned ozone plasma indium-tin-oxide (ITO)-coated glass substrates with a sheet resistance of 20 Ω cm^−2^ and ITO thickness of 100 nm. The hole injection layer (Heraeus Clevios HIL 1.3N; Heraeus Epurio, Leverkusen, Germany/Dayton, OH, USA) was spin-coated (3000 RPM/60 s/acc 800 RPM/s) and annealed onto a hotplate at 200 °C for 3 min to give approximately 50 nm film. The hole transporting layer (PVK) was spun onto the annealed HIL 1.3N layer from 3 mg/mL solution in chloroform: chlorobenzene (95:5 *v*/*v*) (5000 RPM/60 s/acc 600 RPM/s), and then it was annealed at 50 °C for 5 min giving approximately 10 nm film. The emitting layer was spun from a chloroform: chlorobenzene (95:5 *v*/*v*) solution of PVK:PBD (50:50 *w*/*w*) with a total concentration of host 20 mg/mL. The dopant was dissolved in the solution of hosts in order to obtain the final 5% concentration of the emitting layer. The solution was spun (5000 RPM/60 s/acc 600 RPM/s) and then annealed at 50 °C for 5 min giving approximately 70 nm film. All solutions were filtered directly before application using a PVDF or PTFE syringe filter with 0.45 μm pore size. The electron transport layer and cathode layers were thermally evaporated using the Kurt J. Lesker deposition system at 10^−6^ mbar. The electron transport layer (TPBi) and aluminum were deposited at a rate of 1 Å/s. The LiF layer was deposited at a rate of 0.1 Å/s. The characterization of OLED devices was conducted in a 10-inch integrating sphere (Labsphere, North Sutton, NH, USA) connected to a Source Measure Unit (Keithley; Electro Rent, Los Angeles, CA, USA) and coupled with a compact spectrometer AvaSpec ULS2048CL (Avantes, Apeldoorn, The Netherlands).

### 2.2. Synthesis and Characterization

#### 2.2.1. Typical Procedure for the **1**–**4** Synthesis

In a 25 mL flask equipped with a Dean–Stark apparatus were placed 2 mmol (446 mg) of N-ethyl carbazole-3-carboxaldehyde, 2 mmol of adequate cyanoacetate ester (methyl, ethyl, tert-butyl, octyl), 120 μL of acetic acid, 40 mg of dry NH_4_OAc, and 5 mL of dry toluene. The mixture was stirred and heated to reflux for 4 h. The reaction mixture was evaporated to dryness on a rotary evaporator. A solid residue was dissolved in DCM, filtered, and purified using column chromatography (SiO_2_, 2:1 DCM: hexane).

#### 2.2.2. Synthesis of **1**

Synthesized according to the typical procedure. The product was obtained as a yellow solid (yield: 86%). ^1^H NMR (400 MHz, CDCl_3_) δ 8.79 (s, 1H), 8.46 (s, 1H), 8.25 (d, J = 8.7 Hz, 1H), 8.19 (d, J = 7.8 Hz, 1H), 7.57 (t, J = 7.5 Hz, 1H), 7.50 (d, J = 8.7 Hz, 1H), 7.49 (d, J = 7.8 Hz, 1H), 7.36 (t, J = 7.5 Hz, 1H), 4.43 (q, J = 7.2 Hz, 2H), 3.97 (s, 3H), 1.50 (t, J = 7.2 Hz 3H). ^13^C NMR (100 MHz, CDCl_3_) δ 164.20, 156.25, 142.93, 140.68, 129.20, 127.00, 125.48, 123.68, 122.86, 122.62, 121.01, 120.62, 116.96, 109.24, 109.16, 97.25, 53.09, 38.00, 13.85.

#### 2.2.3. Synthesis of **2**

Synthesized according to the typical procedure. The product was obtained as a yellow solid (yield: 79%). ^1^H NMR (400 MHz, CDCl_3_) δ 8.74 (s, 1H), 8.41 (s, 1H), 8.22 (d, J = 8.7 Hz, 1H), 8.17 (d, J = 7.7 Hz, 1H), 7.56 (t, J = 7.7 Hz, 1H), 7.47 (d, J = 7.8 Hz, 1H), 7.45 (d, J = 8.7 Hz, 1H), 7.34 (t, J = 7.6 Hz, 1H), 4.42 (q, J = 7.2, 2H), 4.40 (q, J = 7.3 Hz, 2H), 1.49 (t, J = 7.3 Hz, 3H), 1.44 (t, J = 7.2 Hz, 3H). ^13^C NMR (100 MHz, CDCl_3_) δ 163.66, 155.93, 142.82, 140.64, 129.07, 126.95, 125.39, 123.61, 122.84, 122.65, 120.97, 120.56, 116.96, 109.22, 109.11, 97.73, 62.28, 37.97, 14.30, 13.85.

#### 2.2.4. Synthesis of **3**

Synthesized according to the typical procedure. The product was obtained as a yellow oil, which quickly crystallized (yield: 90%). ^1^H NMR (400 MHz, CDCl_3_) δ 8.70 (s, 1H), 8.33 (s, 1H), 8.20 (dd, J = 8.7 Hz, 1H), 8.16 (d, J = 7.9 Hz, 1H), 7.55 (td, J = 7.4, 1H), 7.46 (d, J = 7.9 Hz, 1H), 7.44 (d, J = 8.7 Hz, 1H), 7.34 (t, J = 7.4 Hz 1H), 4.38 (q, J = 7.2 Hz, 2H), 1.64 (s, 9H), 1.48 (t, J = 7.2 Hz, 3H). ^13^C NMR (100 MHz, CDCl_3_) δ 162.51, 155.03, 142.62, 140.61, 128.87, 126.87, 125.12, 123.54, 122.84, 122.75, 120.93, 120.45, 117.14, 109.18, 109.05, 99.55, 83.05, 37.93, 28.10, 13.85.

#### 2.2.5. Synthesis of **4**

Synthesized according to the typical procedure. The product was obtained as a yellow oil, which crystallized after a few days (yield: 94%). ^1^H NMR (400 MHz, CDCl_3_) δ 8.75 (s, 1H), 8.42 (s, z1H), 8.24 (dd, J = 8.7 Hz, 1H), 8.17 (d, J = 7.8 Hz, 1H), 7.56 (td, J = 7.4 Hz, 1H), 7.47 (d, J = 8.7 Hz, 2H), 7.35 (t, J = 7.4 Hz, 1H), 4.41 (q, J = 7.2 Hz, 2H), 4.35 (t, J = 6.9 Hz, 2H), 1.81 (p, J = 6.9 Hz, 2H), 1.49 (t, J = 7.2 Hz, 3H), 1.51–1.44 (m, 2H), 1.42–1.28 (m, 8H), 0.95–0.89 (m, 3H). ^13^C NMR (100 MHz, CDCl_3_) δ 163.74, 155.88, 142.81, 140.64, 129.05, 126.94, 125.40, 123.61, 122.85, 122.68, 120.97, 120.56, 116.90, 109.22, 109.11, 97.80, 66.42, 37.98, 31.80, 29.22, 29.18, 28.63, 25.86, 22.66, 14.10, 13.85.

## 3. Results and Discussion

It is a well-known fact that the leading cause for this effect is the restriction of intramolecular rotation (RIR) in the aggregates [6]. Our previous paper reports a series of compounds with donor-CH=C(CN)_2_ architecture exhibiting strong AIE behavior. On the other hand, N-ethyl carbazole is an extremely useful “donor” moiety [35]. Thus, we designed and successfully synthesized a series of compounds with N-ethyl carbazole-CH=C(CN)(COOR) architecture. In this regard, we achieved molecules with the exact energetic parameters (HOMO and LUMO level, bandgap, etc.) but with different possibilities of intermolecular interactions due to alkyl chains with different lengths and types (R substituent in the ester group). For clarity, the structures of the obtained and evaluated compounds are presented in Figure 1.

### 3.1. DFT Calculations

In the first stage of the research, detailed DFT calculations were performed. It was crucial to confirm our assumption that exchanging the substituent in the ester group (marked in red in Figure 1) has a negligible influence on single-molecule energetic (HOMO, LUMO, IP, EA, band gap), electrochemical, and optical properties. For this purpose, the Gaussian16 program was used [36]. Simulated geometries of the investigated compounds were optimized in a vacuum using B3LYP/6-311+G. Based on these geometries, molecular frontier orbitals were calculated in MeCN (acetonitrile).

Discussed molecules are based on the Donor-π-Acceptor (D-π-A) structure (where the donor is N-ethyl-carbazol-3-yl substituent, while the acceptor group is a 2-cyanoacrylic ester moiety). Therefore, it was decided that the contributions of individual parts of the frontier molecular orbital be analyzed. As expected, both HOMO and LUMO levels are similar for all molecules (see Table 1). As expected, the HOMO orbital is located mainly on the carbazole part [37]. On the other hand, the LUMO orbital is more dominated by the contribution from 2-cyanoacrylic ester moiety. The IP, EA, and energy bandgap (Eg(DFT)) were also calculated to check the effect of a 2-cyanoacrylic ester derivative moiety change on the properties of the compounds (Table 2). For all molecules, the IP values are almost the same (i.e., between −5.79 and −5.85 [eV]). In the case of EA, values are also similar and are in the range of −2.98 to −3.06 [eV]. Summing up the parameters mentioned above, the investigated compound series’ energy gap was 2.78–2.84 [eV]. However, this indicates that the acceptor group’s modification does not cause significant changes in the energy gap width.

Additionally, bond lengths for each of the investigated compounds were determined by DFT calculation. The first crucial observation is that the bond between carbazole and vinyl moieties is shorter than the bonds within carbazole itself. It proves high electron density flow from donor to acceptor. Moreover, all bonds in the Acceptor part of the molecule (i.e., CH=C(CN)(COOR)) are almost equal, and they are intermediate between a single and a double bond, which implies good resonance contact through a hole molecule and good charge mobility.

### 3.2. Photophysical Characterization

Considering the results of the DFT calculations, we turn our attention to the photophysical characterization of the investigating compounds by electrochemical (Figure 2) and optical measurements (Figure 3). The E_red_ (peak onset) value and the shape of the reduction wave suggest that the reduction occurs on the acceptor fragment (similar behavior was observed for carbazole-CH=C(CN)_2_ compound [29]). Moreover, the value of the reduction potentials for the diluted solution is always similar. In turn, oxidation always occurs within the π-excising carbazole moiety (a curve characteristic for irreversible carbazole/carbazole^+^ oxidation was obtained). However, the crucial aspect is that, in diluted solutions, the IP, EA, and electrochemical bandgap are almost equal (differences below 10 eV, see Table 2 and Figure 2). As shown in Figure 3, the optical band gaps in diluted solutions are also very similar. Each of the compounds exhibit two bands within the available region: one with a maximum peak of 327 nm (approximately) and the second one at 400 nm, while peak onsets are as high as 445 nm (±2 nm). At this research stage, we have also evaluated the influence of the f_w_ (water fraction) on absorption. As one can see in each case in the water, a blue shift of the bandgap was observed in the MeCN mixture (9:1 *v/v*, water served as “poor” solvent while MeCN was “good” solvent). This means that in solutions forcing aggregation, stabilization energy occurs due to the intermolecular interactions. To better understand this phenomenon, a series of TD-DFT calculations were performed using the same function and base as mentioned above. Both experimental and theoretical spectra are summarized in Appendix A. All compounds absorb light in a wide band with an absorption maximum of around 390 nm. This band corresponds to the HOMO-LUMO transitions (where HOMOs have 80–86% donor and 14–20% acceptor contributions, while the LUMOs consist of 37–48% of donors and 52–63% of acceptors). The performed calculations indicate that in the discussed band we also deal with the H-1-LUMO transitions (where HOMO-1 has 95–97% donor and 3–5% acceptor contributions). The oscillator strength for these transitions correspond to HOMO→LUMO above 0.47 and H-1→LUMO above 0.24. As can be seen, the experimental results are in good agreement with the calculations.

### 3.3. Aggregation-Induced Emission Investigations

As mentioned in the introduction, PL yields in dilute solution are poor for this type of compound. However, as we reported previously, for similar compounds, significant enhancement of the quantum yields in the solid was recorded and are strongly hypsochromically shifted, suggesting the existence of AIE properties. Thus, the study of the AIE properties was crucial for compounds designed and synthesized within this research. After preliminary tests, we selected the mixture of MeCN (as a good solvent) and water (as a poor solvent) with different fractions of water (fw) ranging from 0% to 99%. As one can see in the photos presented in Figure 4, compounds exhibit typical AIE behavior (for clarity, only compounds **1** and **3** are given; for others, see Appendix A). For the studied compounds in dilute MeCN solution (10^−5^ mol/L), almost no fluorescence was observed under a 366 nm UV lamp. Furthermore, no apparent changes were observed when the water: MeCN ratio was below 60%. However, when fw > 90%, outstanding emission enhancement was observed for **1**. This behavior can be attributed to π-π stacking aggregation resulting in the blocking of the nonradioactive processes. However, the essential part of this research was the comparison of the compounds with different substituents. As one can see with even the naked eye comparing photo A (**1**, i.e., compound with small Me substituent) with photo C (**3,** i.e., the compound with tert-butyl as the substituent), differences are spectacular. In the case of compound **3**, the AIE feature also occurs but is visibly weaker. Secondly, light was emitted during exposure to a UV-vis lamp (366 nm, see photos A and C in Figure 4). Irradiation is also different, proving that intermolecular interaction plays a crucial role. Considering that aggregation is time-dependent (which is sometimes not taken into consideration during research, unfortunately), all measurements and observations were conducted twice—immediately after solution preparation and after 24 h. As one can see in photo B (Figure 4) for the solutions I–VII (i.e., for compound **1** in mixtures containing > 80% of water), the differences are almost negligible—the solution is stable, and no aggregation occurs, except for in the case of solutions with higher water fraction suspension with high PL yield precipitated. Moreover, we have observed that the precipitated solid emits light with different wavelengths for a solution with fw as high as 99%. This is because, at high water content, aggregation is forced by poor solvent and is too fast—thus, the molecules cannot achieve the most favorable configuration. On the other hand, the similar behavior of solution **3** occurs on a much smaller scale. This means that an expanded substituent effectively prevents intermolecular interactions.

As mentioned earlier, during the preliminary research several measurements of the impact of the aggregation degree on the emissions quantum yields (depending on the structure of the molecule) were also made. In the case of compound **1**, a small methyl substituent was introduced into the ester structure. It should have allowed the molecule to achieve a planar configuration and not hinder aggregation. On the other hand, **4** possesses a long chain (octyl) that should impede aggregation. In the case of the **1** molecule in dilute solution (10^−5^ mol/L in MeCN), the maximum emission occurs at 495 nm regardless of the excitation wavelength (Figure 5) with almost equal efficiency of about 3%. For the concentrated solution (10^−3^ mol/L in MeCN), an increase in quantum efficiency up to 3.5% (λex = 438) was observed. However, a lowering of the emissions quantum yields for excitation with other wavelengths occurred (Figure 6). Furthermore, what is worth noting for a solution in a MeCN: water mixture (1:9) (with Cm = 10^−5^ mol/L) is that a spectacular increase in quantum yield (up to 89%!) takes place (Figure 7).

Interestingly, despite using different λex, each case’s emission band has two maxima. Surprisingly, by changing the methyl with the ethyl group, the PL yield drops almost four times. A similar procedure was performed for the other molecules and is summarized in Table 3. In the case of **4**, going from the diluted MeCN solution to the MeCN: water mixture, an increase in quantum yield was also observed but was significantly smaller, i.e., from 3% up to 14%. This results from difficulties in intermolecular interactions due to a long alkyl chain. However, the weaker AIE behavior was detected for compounds possessing tert-butyl moiety. This means that the tert-butyl group makes intermolecular interaction the most difficult.

However, the best summary of the conducted research is the photos presented in Figure 8. The pictures were taken by placing the camera inside the fluorimeter chamber during measurements related to absorptions λ_max_. As one can see in the case of the diluted solutions in MeCN (10^−5^ mol/L), each compound’s behavior is similar. However, it is worth noting that differences are significant for mixtures of MeCN with water. The most intense light emission was observed for **1** (see Table 3), while for **3** and **4,** emitted light was visibly weaker.

Finally, the OLEDs were fabricated by the spin-coating/evaporation hybrid method. The hole injection layer (Heraeus Clevios HIL 1.3N), hole transporting layer (PVK), and emitting layer (PVK:PBD + dopant) were spin-coated, whereas the electron transport layer (TPBi) and cathode (LiF/Al) were evaporated. The overall structure of the device was ITO | HIL 1.3N (50 nm) | PVK (10 nm) | PVK:PBD (50:50) with 5 mol% **1**–**4** (70 nm) | TPBi (50 nm) | LiF (1 nm) | Al (100 nm). Devices showed green–blueish electroluminescence (Figure 9) with CIE coordinates: 0.107, 0.165 (see Figure 9).

The obtained OLEDs showed relatively high turn-on voltage V_ON_ 12–14 V at 1.5 cd m^−2^, and maximum luminance reached 25 cd m^−2^ (see Figure 10). Modest device luminance can be attributed to relatively low current density, which was below 30 mA cm^−2^. The EQE (ratio of the number of photons produced per recombined electron–hole), was rather satisfactory, reaching approximately 2% which is quite a common result for LEDs made with the spin-coating method. Moreover, it should be noted that the above results refer to the first trials without optimizing the OLED structure. Thus, the proposed molecular architecture is extremely promising for the further development of organic electronics.

## 4. Conclusions

During AIE research, a group of compounds with identical energy parameters was analyzed (energy levels were determined using both cyclic voltammetry and UV-vis spectroscopy for diluted solutions). HOMO and LUMO energy levels of studied compounds exhibit almost no difference. This results in practically identical behavior of these substances in dilute solutions—it has been proven that both the reduction and oxidation potentials, as well as the optical properties, are similar. On the other hand, for molecules that exhibit the strong AIE phenomenon, emission efficiency increases rapidly during aggregation. What is also very interesting is it has been shown that by introducing appropriate substituents, one can control the degree of intermolecular interactions and “control” the length of the emitted wave. Moreover, the time dependence on the photophysical properties of the molecules exhibiting AIE features was confirmed and evaluated. Finally, OLEDs were fabricated by the spin-coating/evaporation hybrid method. Devices showed green–blueish electroluminescence (CIE coordinates: 0.107, 0.165) with maximum luminance reaching 25 cd m^−2^ and reaching approximately 2% of EQE. The results confirm that the proposed molecular architecture is promising for developing compounds with a high AIE index. The presented synthesis strategy allows for obtaining pure compounds with the expected photophysical properties in a cheap, convenient, and large-scale manner.

## Figures and Tables

**Figure 1 materials-15-08525-f001:**
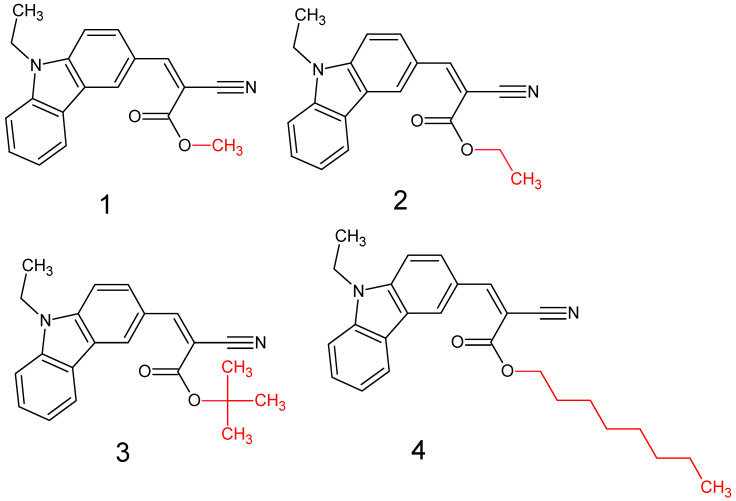
Structures of obtained compounds (**1**–**4**).

**Figure 2 materials-15-08525-f002:**
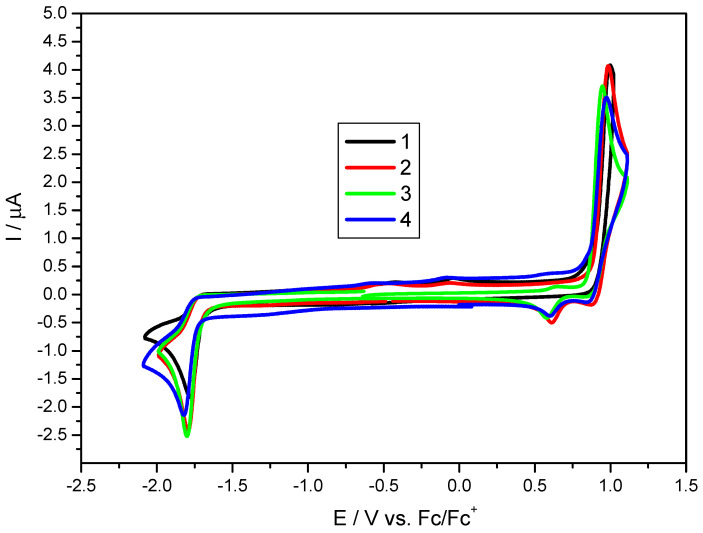
Cyclic voltammograms of the investigated compounds with sweep rate ν = 100 mV/s, 0.1 M Bu_4_NPF_6_ in MeCN.

**Figure 3 materials-15-08525-f003:**
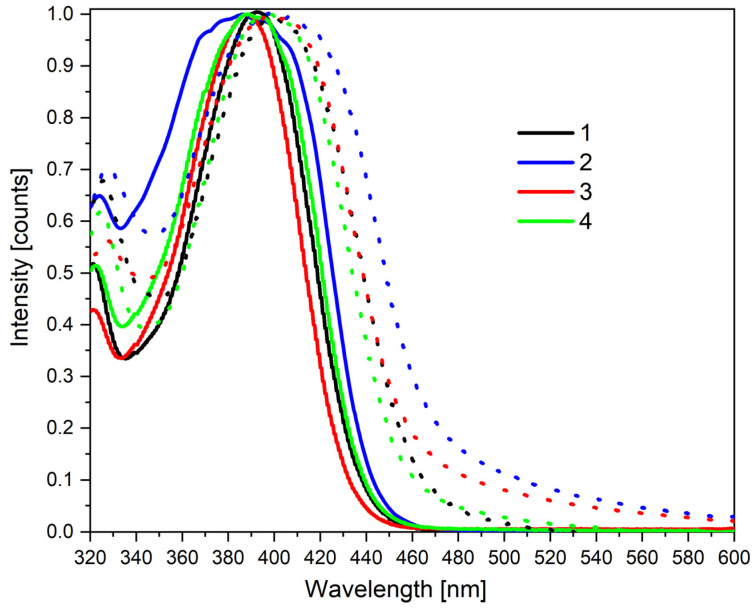
Normalized electronic absorption in MeCN c = 1 × 10^−5^ mol/L (solid lines) and MeCN/water mixture (c = 1 × 10^−5^ mol/L) 9:1 *v/v* (dotted lines).

**Figure 4 materials-15-08525-f004:**
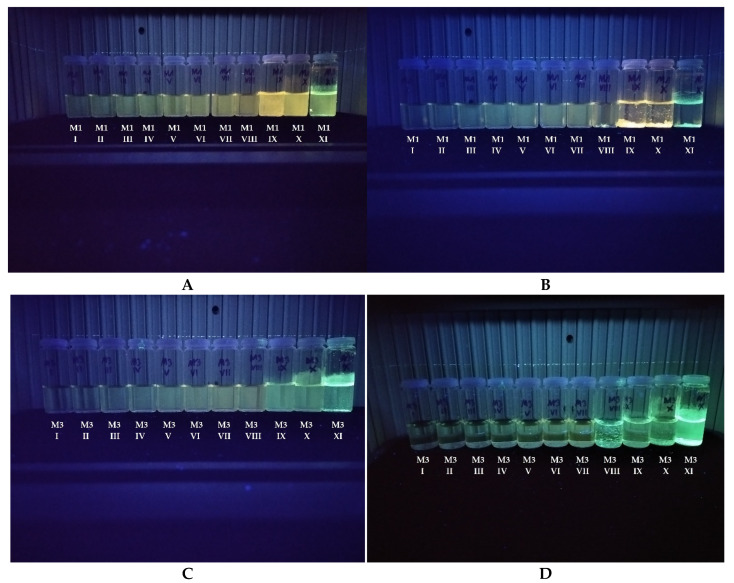
Photos taken during illumination by light with 366 nm wavelength. In each column, the vessels are arranged to increase water fraction in water: MeCN solutions (from left to right), i.e., I. 10%; II. 20%; III. 30%; IV. 40%; V. 50%; VI. 60%; VII. 70%; VIII. 80%; IX. 90%; X. 95%; XI. 99%. (**A**) “Fresh” **1** solution; (**B**) solutions of **1** after 24 h. (**C**) “Fresh” **3** solutions; (**D**) solutions of **3** after 24 h.

**Figure 5 materials-15-08525-f005:**
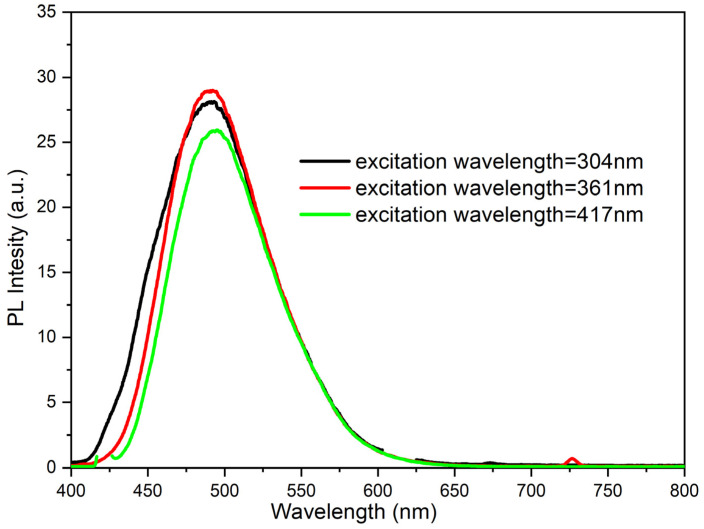
PL spectra of compound **1** in dilute solution (10^−5^ mol/L in MeCN).

**Figure 6 materials-15-08525-f006:**
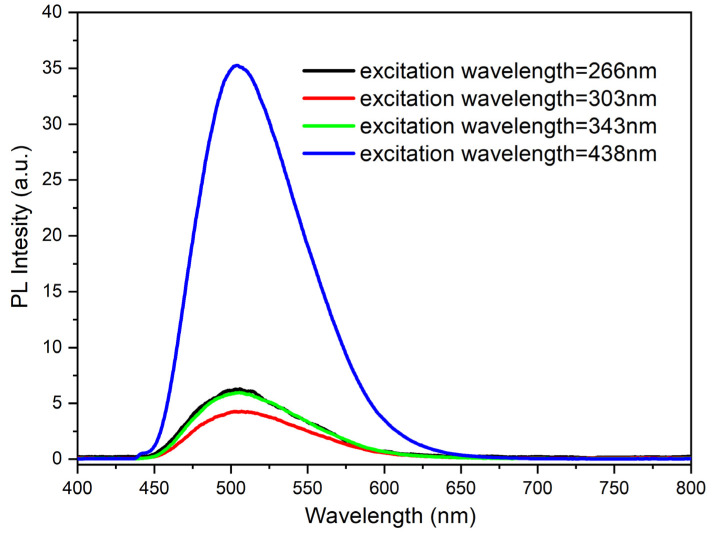
PL spectra of compound **1** in solution (10^−3^ mol/L in MeCN).

**Figure 7 materials-15-08525-f007:**
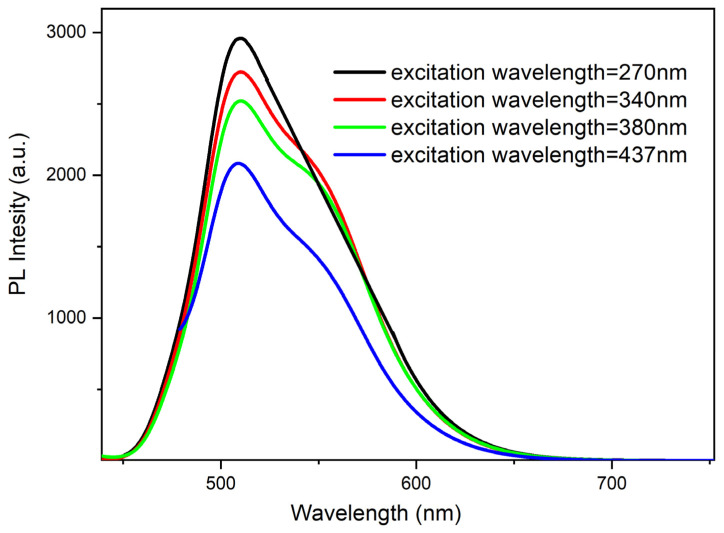
PL spectra of compound **1** in water/MeCN mixture 9:1 solution (10^−5^ mol/L).

**Figure 8 materials-15-08525-f008:**
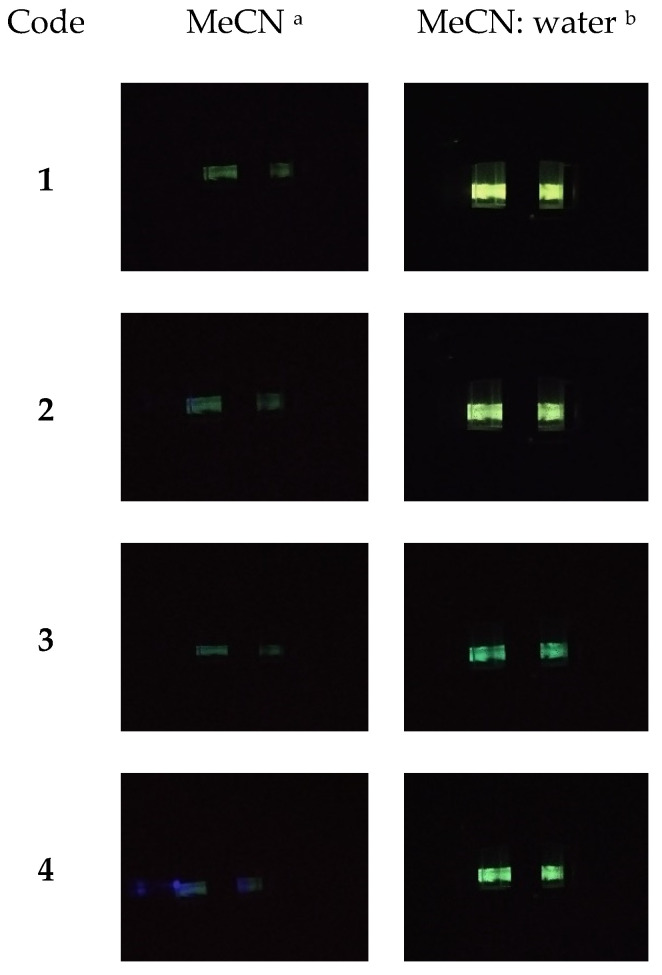
Pictures of investigated compounds from inside the fluorescence spectrophotometer during irradiation (excitation wavelengths similar to those reported in Table 3). ^a^ (c = 1 × 10−5 mol/L), ^b^ (c = 1 × 10^−5^ mol/L) 9:1 *v/v*.

**Figure 9 materials-15-08525-f009:**
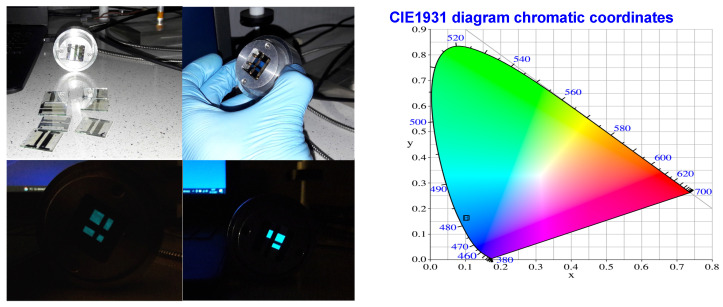
Pictures of OLED devices, based on compound **2,** synthesized during preliminary tests together with color coordinates.

**Figure 10 materials-15-08525-f010:**
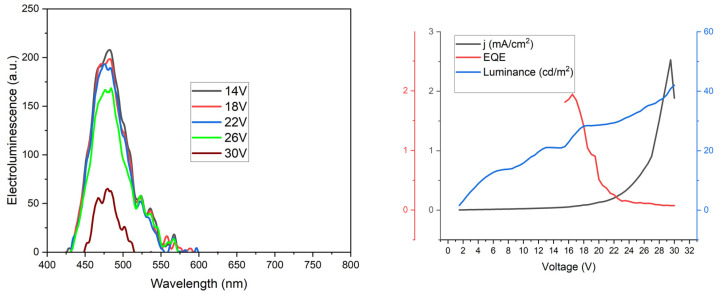
Electroluminescence spectrum (left) and current density, EQE–luminance–voltage curves for tested OLED device (2 × 2 mm). The overall structure of the device was ITO | HIL 1.3N (50 nm) | PVK (10 nm) | PVK:PBD (50:50) with 5 mol% **2** (70 nm) | TPBi (50 nm) | LiF (1 nm) | Al (100 nm).

**Table 1 materials-15-08525-t001:** The HOMO and LUMO levels, together with calculated bond lengths.

CODE	HOMO	LUMO	BOND LENGTHS
1	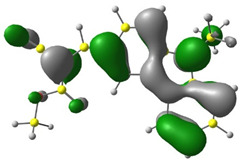	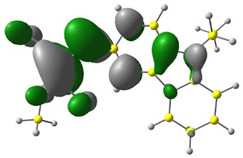	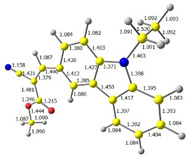
	E = −6.03 [eV]	E = −2.66 [eV]	
2	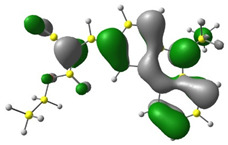	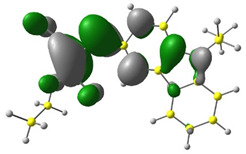	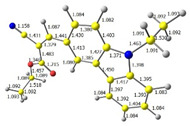
	E = −6.03 [eV]	E = −2.64 [eV]	
3	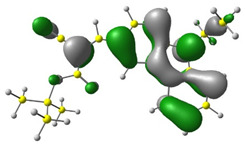	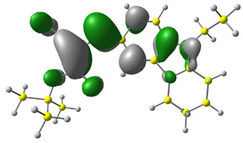	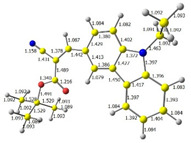
	E = −6.00 [eV]	E = −2.59 [eV]	
4	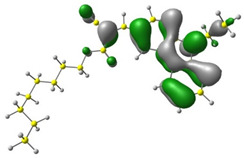	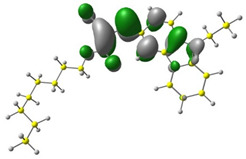	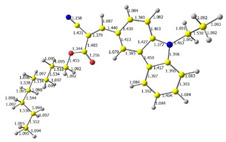
	E = −6.02 [eV]	E = −2.64 [eV]	

**Table 2 materials-15-08525-t002:** IP, EA, and energy gap values for **1**–**4**.

Code	Eox [V]	Ered [V]	IP (CV) ^a^ [eV]	EA (CV) ^a^ [eV]	Eg (CV) ^a^ [eV]	Eg (Opt) ^b^ [eV]	IP (DFT) ^c^ [eV]	EA (DFT) ^c^ [eV]	Eg (DFT) ^c^ [eV]
**1**	0.89	−1.71	−5.99	−3.39	2.6	2.83	−5.84	−3.00	2.84
**2**	0.88	−1.72	−5.98	−3.38	2.6	2.8	−5.82	−3.03	2.80
**3**	0.86	−1.72	−5.96	−3.38	2.58	2.87	−5.80	−2.98	2.82
**4**	0.87	−1.74	−5.97	−3.36	2.61	2.82	−5.84	−3.02	2.81

^a^ Calculated from CV measurements (IP = −5.1 − E_ox_; EA = −(5.1 − E_red_)); Eg _(CV)_ = E_ox (onset)_ − E_red (onset))_; ^b^ obtained from UV-Vis using Eg = 1240/λ_onset_ formula; ^c^ obtained from DFT.

**Table 3 materials-15-08525-t003:** Photophysical parameters obtained from UV-vis spectroscopy and photoluminescence measurements.

Code	UV-Vis Absorption λ_max_ [nm]			PL (Photoluminescence) λ_em_ [nm]		
	MeCN ^a^	MeCN ^b^	MeCN: Water ^c^	MeCN ^a^ (Φf)	MeCN ^b^ (Φf)	MeCN: Water ^c^ (Φf)
M1	304; 364; **417**	266; 303; 343; **438**	270; 340; **380**; 437	495 (3%)	504 (3%)	510 (89%)
M2	267; 275; 304; 364; **410**	260; 300; 337; **445**	234; 301; **365**	495 (3%)	505 (3%)	504 (78%)
M3	272; 304; 344; **429**	267; 304; 324; **447**	210; **376**; 410	506 (3%)	500 (3%)	515 (9%)
M4	268; 299; 350; **423**	260; 297; 330; **439**	**273**; 305; 359	500 (3%)	505 (3%)	515 (14%)

Φf—quantum yield determined using a calibrated integrating sphere, in bold—wavelength used for excitation in PL measurements; ^a^ (c = 1 × 10^−5^ mol/L) ^b^ (c = 1 × 10^−3^ mol/L) ^c^ (c = 1 × 10^−5^ mol/L) 9:1 *v/v*.

## Data Availability

Not applicable.

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
