# Peer review of "The Role of Intermolecular Interaction on Aggregation-Induced Emission Phenomenon and OLED Performance"

_materials, 2022, doi:10.3390/ma15238525_

Round 1

Reviewer 1 Report

In this manuscript, Filipek et al. studied the effects of intermolecular interaction on aggregation-induced emission for OLED applications. Four different side groups (methyl, ethyl, tert-butyl, and octyl) were attached to the light-emitting molecule, and their emission properties were investigated via theoretical and experimental methods. Overall, the manuscript is well-organized with sufficient data. However, before consideration of this manuscript for publication, the following issues should be resolved.

(1) The OLED characteristics in Figure 10 are not trustworthy. The EQE higher than 75% is unconventionally high, and it was obtained at a bias much lower than the turn-on voltage. Further, the light emission of the OLED was observed at a very lower bias than the turn-on voltage. These results are not observed in other reports. In addition, although the authors described that the maximum luminance of 3500 cd/m2 was obtained, no data are in Figure 10. The authors must check the reliability of the data and rewrite the device result section. Also, the authors are requested to add how they calculated the EQE in this study.

(2) The molecular structures in the synthesis section should be removed because they are also shown in Figure 1.

(3) In Table 1, the HOMO, LUMO, and bond lengths of (1, 2) and (3, 4) are drawn with the opposite view: carbazole moiety is located on the left side in (1, 2) whereas on the right side in (3, 4). For a clear comparison, they should be shown in the same view.  

(4) Please insert the unit [eV] in the all parameter names, like IP [eV], EA [eV], and Eg [eV] in Table 2. For a clear comparison, IP and EA should be IP(CV), EA(CV), Eg(cv) should be Eg(CV), and Eg should be Eg(DFT).

(5) The sample names in Figure 4 are dimly visible. The authors are suggested to add the sample name in the figure.

(6) Many minor errors were observed:

[line 57, page 2 and line 61, page] Please unify the reference nomenclature: “Li et al.” or “Hui Li et al.” or “H. Li et al.”.

[line 98, page 3 et al.] “app” should be “approximately”,

[line 98, page 3] “PVKH” should be “PVK”,

[line 186, page 5 and line 228, page 7 et al.] Please unify the name of “acetonitrile” and “MeCN”,

and so on. The manuscript should be revised thoroughly. 

Author Response

Dear Reviewer

We thank You for Your detailed comments on our manuscript "The role of intermolecular interaction on aggregation-induced emission phenomenon and OLED performance" (materials-2023954)

We have carefully considered all of the comments. Modifications and corrections have been added accordingly in reply to the suggestions. Please find below enclosed our detailed point-by-point answers [modifications introduced into paper appearing in blue ink] to the comments [appearing in italic black ink].

Answer to comments of Reviewer #1:

Remark 1:

The OLED characteristics in Figure 10 are not trustworthy. The EQE higher than 75% is unconventionally high, and it was obtained at a bias much lower than the turn-on voltage. Further, the light emission of the OLED was observed at a very lower bias than the turn-on voltage. These results are not observed in other reports. In addition, although the authors described that the maximum luminance of 3500 cd/m2 was obtained, no data are in Figure 10. The authors must check the reliability of the data and rewrite the device result section. Also, the authors are requested to add how they calculated the EQE in this study.

Answer:

We agree that EQE was unconventionally high. This yield was determined for a very low current, which has no practical significance. Moreover, we have also found that there was some error in the reference file that was used for all calculations; therefore, data were recalculated for current values that induce a stable luminescence visible even to the naked eye. The text in the manuscript was corrected.

Obtained OLEDs showed relatively high turnon voltage VON 12-14 V at 1,5 cd m-2, and maximum luminance reaching 25 cd m-2.Modest device luminance can be attributed to relatively low current density, which was below 30 mA cm-2. The EQE (ratio of a number of photons produced per recombined electron-hole), was rather low, reaching max app 2%, which is quite a common result for LEDs produced with the spin coating method. Moreover, it should be noted that the above results refer to the first trials, without optimizing the OLED structure. Thus, the proposed molecular architecture is extremely promising for the further development of organic electronics.

The EQE calculation is a multi-step process and is done by the program. In simplified terms, EQE is counted as the ratio of number of photons produced per recombined electron-hole pair ((Photons emitted)/ (charge carriers injected)).

The amount of charge carriers injected is calculated by dividing the injected charge by the charge of the electron. The charge is taken from the current density, taking into account the area and time (750 ms). The number of generated photons per time unit is converted from the EL data. Measurement data (detector current) is compared with a reference file which is generated on the basis of a calibrated light source. The obtained luminous power is divided by the energy corresponding to the maximum emission, yielding the number of photons.

Remark 2:

The molecular structures in the synthesis section should be removed because they are also shown in Figure 1.

Answer:

Thank you for your notice. The manuscript was corrected according to the Reviewers suggestion.

Remark 3:

In Table 1, the HOMO, LUMO, and bond lengths of (1, 2) and (3, 4) are drawn with the opposite view: carbazole moiety is located on the left side in (1, 2) whereas on the right side in (3, 4). For a clear comparison, they should be shown in the same view.  

Answer:

Pictures have been standardized. All molecules are shown with the view presenting carbazole to the right side for better comparison.

Remark 4:

Please insert the unit [eV] in the all parameter names, like IP [eV], EA [eV], and Eg [eV] in Table 2. For a clear comparison, IP and EA should be IP(CV), EA(CV), Eg(cv) should be Eg(CV), and Eg should be Eg(DFT).

Answer: Thank you for your remark. The manuscript was corrected according to the Reviewer's suggestion.

Remark 5:

The sample names in Figure 4 are dimly visible. The authors are suggested to add the sample name in the figure.

Answer:

Thank you for pointing this out. We added the sample name to the figure.

Remark 6:

            Many minor errors were observed:

[line 57, page 2 and line 61, page] Please unify the reference nomenclature: "Li et al." or "Hui Li et al." or "H. Li et al.".

[line 98, page 3 et al.] "app" should be "approximately",

[line 98, page 3] "PVKH" should be "PVK",

[line 186, page 5 and line 228, page 7 et al.] Please unify the name of "acetonitrile" and "MeCN",  and so on. The manuscript should be revised thoroughly. 

Answer:

Thank you for your notice. We apologize for all these mistakes. The manuscript was reviewed several times and corrected.       

Reviewer 2 Report

MeCN - even though it is easy to guess what it is, please indicate clearly what it is.

Please correct the word "Wavelength" - I found "Wavelenght" all over the manuscript.

Fig.3 - can you explain why the electronic absorption in MeCN and MeCN/water mixture of the compound 2 is much wider than of others compounds? Moreover, the curves are more "dotted" ones than "dashed" ones.

It is difficult to figure out which final composition you used in the OLED structure (it would be interesting to indicate it in the photos). What is the lifetime of these devices? Did you perform the accelarated lifetime tests?

Why a part of the spectra in the fig5S is missing?

There are some strange pics observed, like in Fig 7S (not only in this one, but I take is as an example) - black curve, at about 535 nm or red curve at about 603 nm - these are 2nd orders of the excitation wavelengths? Indicate it in the paper and comment please.

Please unify the references (sometimes the names are after or before the surnames, there is a problem with the name in ref. 7).

Author Response

Dear Reviewer

We thank You for Your detailed comments on our manuscript "The role of intermolecular interaction on aggregation-induced emission phenomenon and OLED performance" (materials-2023954)

We have carefully considered all of the comments. Modifications and corrections have been added accordingly in reply to the suggestions. Please find below enclosed our detailed point-by-point answers [modifications introduced into paper appearing in blue ink] to the comments [appearing in italic black ink].

Answer to comments of Reviewer #2:

Remark 1:

MeCN - even though it is easy to guess what it is, please indicate clearly what it is.

Answer:

            Thank you for your remark. The manuscript was corrected according to the Reviewer suggestion.

Remark 2:

Please correct the word "Wavelength" - I found "Wavelenght" all over the manuscript.

Answer:

               Thank you for this remark. We apologize for this mistake. The word "Wavelenght" was inside the figure, so the typo was not noticed during the text correction.

Remark 3:

Fig.3 - can you explain why the electronic absorption in MeCN and MeCN/water mixture of the compound 2 is much wider than of others compounds? Moreover, the curves are more "dotted" ones than "dashed" ones.

Answer:

Thank you for your notice. The Figure caption was corrected. In our opinion in the case of 2 absorption peak is much wider due to the intermolecular interaction causing the formation of various type of aggregates in the solution.

Remark 4:

It is difficult to figure out which final composition you used in the OLED structure (it would be interesting to indicate it in the photos). What is the lifetime of these devices? Did you perform the accelarated lifetime tests?

Answer:

Thank you for this remark. The final composition of the OLED structure was added to the Figure caption. We have not performed lifetime measurements yet. However, the device emits light steadily for several minutes, which suggests that their lifetime is rather satisfactory.

Remark 5:

Why a part of the spectra in the fig5S is missing?

Answer:

Thank you for your notice. Measurements were have been repeated, and figure 5S was corrected.

Remark 6:

There are some strange pics observed, like in Fig 7S (not only in this one, but I take is as an example) - black curve, at about 535 nm or red curve at about 603 nm - these are 2nd orders of the excitation wavelengths? Indicate it in the paper and comment please.

Answer:

Thank you for your remark. The Reviewer's observation is correct. Figures in supporting information  have been corrected.

Remark 7:

Please unify the references (sometimes the names are after or before the surnames, there is a problem with the name in ref. 7).

Answer:

We are sorry for all these mistakes. As the Reviewer suggested, the appropriate correction was included in the revised version of the article.

Round 2

Reviewer 1 Report

The manuscript has been improved after revision. However, the description of the L of 3500 cm/m2 is still present in the abstract and conclusion, but it cannot be seen in Figure 10. If this error is corrected, I agree with the publication of this manuscript in Materials.  

Author Response

Dear Reviewer

We thank You for Your detailed comments on our manuscript “The role of intermolecular interaction on aggregation-induced emission phenomenon and OLED performance” (materials-2023954)

Answer to comment:

Remark 1:

The manuscript has been improved after revision. However, the description of the L of 3500 cm/m2 is still present in the abstract and conclusion, but it cannot be seen in Figure 10. If this error is corrected, I agree with the publication of this manuscript in Materials.  

Answer:

Thank you for your notice. The manuscript was corrected according to the suggestion.

Reviewer 2 Report

Page 2, line 62 "-CH=C(CN)2", make 2 be an index

Please add the explanation of the "MeCN" abbreviation.

Please explain the phrase in the page 7 "During electrochemical experiments, it has 220 been shown that the reduction occurs exclusively on acceptor fragments."

please explain the phase in the page 7 "In turn, oxidation always takes place 222 within the π-excising carbazole moiety."

Page 7 line 224 - eV does not need to be in square brackets.

All compounds absorb light in a wide band with an absorption maximum of around 390 nm. It specifies the π-π * transitions. - I understand what is meant, but "it specifies the π-π * transitions." sounds very strange. It is better to say " the π-π * transition is responsible for this absorption." or something like that.

Indicate what are the wavelengths given in the UV-Vis λmax [nm] columns of the Table 3. Is it absorption?

Frankly speaking, with the 2% of  EQE, to say in the conclusions that "The results confirm that the proposed molecular architecture is extremely promising for developing organics electronics." is far too optimistinc...

Author Response

Reviewer 2:

Dear Reviewer

We thank You for Your detailed comments on our manuscript “The role of intermolecular interaction on aggregation-induced emission phenomenon and OLED performance” (materials-2023954)

Remark 1:

“Page 2, line 62 “-CH=C(CN)2”, make 2 be an index”; “Please add the explanation of the “MeCN” abbreviation.”; “Indicate what are the wavelengths given in the UV-Vis λmax [nm] columns of the Table 3. Is it absorption?” and “Page 7 line 224 - eV does not need to be in square brackets.”

Answer:

Thank you for your remark. The manuscript was corrected according to the Reviewer’s suggestion.

Remark 2:

Please explain the phrase in the page 7 “During electrochemical experiments, it has 220 been shown that the reduction occurs exclusively on acceptor fragments.” AND

please explain the phase in the page 7 “In turn, oxidation always takes place 222 within the π-excising carbazole moiety.”

Answer:

Thank you for pointing this out. The manuscript was rewritten:

(…) The Ered (peak onset) value and the shape of the reduction wave suggest that the reduction occurs on the acceptor fragment (similar behavior was observed for carbazole-CH=C(CN)2 compound [30]). Moreover, the value of the reduction potentials for the diluted solution is always similar. In turn, oxidation always occurs within the π-excising carbazole moiety (a curve characteristic for irreversible carbazole/carbazole+ oxidation was obtained). (…)

Remark 3:

All compounds absorb light in a wide band with an absorption maximum of around 390 nm. It specifies the π-π * transitions. - I understand what is meant, but “it specifies the π-π * transitions.” sounds very strange. It is better to say “the π-π * transition is responsible for this absorption.” or something like that.

Answer:

 Thank you for your notice. The manuscript was corrected:

(…) All compounds absorb light in a wide band with an absorption maximum of around 390nm. This band corresponds to the HOMO - LUMO transitions (where: HOMOs have 80-86 % donor and 14-20 % acceptor contributions, while the LUMOs consist of 37-48 % of donors and 52-63 % of acceptors). (…)

Remark 4:

 Frankly speaking, with the 2% of  EQE, to say in the conclusions that “The results confirm that the proposed molecular architecture is extremely promising for developing organics electronics.” is far too optimistinc...

Answer:

Thank you for your remark. The conclusions were changed to:

(…) The results confirm that the proposed molecular architecture is promising for developing compounds with a high AIE index. The presented synthesis strategy allows for obtaining pure compounds with the expected photophysical properties in a cheap, convenient, and large-scale manner.